# Heavy Flavor Physics at the sPHENIX Experiment

Zhaozhong Shi [†] on behalf of the sPHENIX Collaboration

Los Alamos National Laboratory, Los Alamos, NM 87545, USA; zhaozhongshi@lanl.gov; Tel.: +1-415-350-3181
[†] Current address: Brookhaven National Laboratory, Upton, NY 11973, USA .

**Abstract:** The sPHENIX experiment is a state-of-the-art jet and heavy flavor physics detector, which successfully recorded its first Au + Au collision data at 200 GeV at the Relativistic Heavy Ion Collider (RHIC). sPHENIX will provide heavy flavor physics measurements at RHIC, covering an unexplored kinematic region and unprecedented precision, to probe the parton energy loss mechanism, parton transport coefficients in quark–gluon plasma, and the hadronization process under various medium conditions. At the center of sPHENIX, the monolithic active pixel sensor (MAPS)-based VerTeX detector (MVTX) is a high-precision silicon pixel detector. The MVTX provides excellent position resolution and the capability of operating in continuous streaming readout mode, allowing precise vertex determination and recording a large data sample, both of which are particularly crucial for heavy flavor physics measurements. In this work, we will show the general performance of heavy-flavor hadron reconstruction. In addition, we will discuss the commissioning experience with sPHENIX. Finally, we will provide the projection of b-hadron and jet observables and discuss the estimated constraints on theoretical models.

**Keywords:** sPHENIX; heavy-ion collisions; quark–gluon plasma; heavy flavor physics; particle reconstruction; commissioning; data taking; silicon pixel detector; vertexing; streaming readout; cosmic data; b-hadrons

## 1. Introduction

The sPHENIX experiment [1] is the first new experiment at RHIC in over 20 years. According to the 2015 [2] and 2023 [3] United States Nuclear Science Advisory Committee (NSAC) Long Range Plan for Nuclear Science, the sPHENIX is considered as a Department of Energy flagship experiment in heavy ion physics. The physics goal of sPHENIX is to probe the inner workings of quark–gluon plasma (QGP) by resolving its properties at shorter and shorter length scales, and sPHENIX is complementary with experiments at the LHC.

sPHENIX is a state-of-the-art heavy flavor and jet detector at RHIC. It has a $2\pi$ angular acceptance over the rapidity range of $|y| < 1$. It consists of tracking and calorimeter systems, with excellent capabilities for studying the strongly interacting QCD matter at RHIC. In addition, sPHENIX is equipped with a minimum bias detector (MBD) and a zero degree calorimeter (ZDC) in the far forward region and a trigger system for global event characterization. A schematic drawing of the sPHENIX detector is shown in Figure 1 below.

As the inner tracking system, the monolithic active pixel sensor (MAPS)-based VerTeX detector (MVTX) and the intermediate silicon strip tracker (INTT) leverage advanced silicon detector technologies. The outer tracking system is made of a compact time projection chamber (TPC) and the TPC outer tracker (TPOT). The sPHENIX experiment, complementary to the LHC experiments for heavy-ion physics studies, is equipped with a tracking system, like the ALICE inner tracking system-2 (ITS-2) at the LHC, capable of operating in both triggered and continuous streaming readout modes. The streaming readout acts as a triggerless configuration, capable of sending all collisions to the data acquisition system. The INTT provides us with a timing resolution of approximately 100 ns, which can resolve RHIC bunch crossings. By utilizing the ACTS tracking algorithm with excellent tracking

purity and efficiency [4], and thanks to the excellent performance of the tracking detectors, sPHENIX achieves precision momentum resolution and vertexing determination.

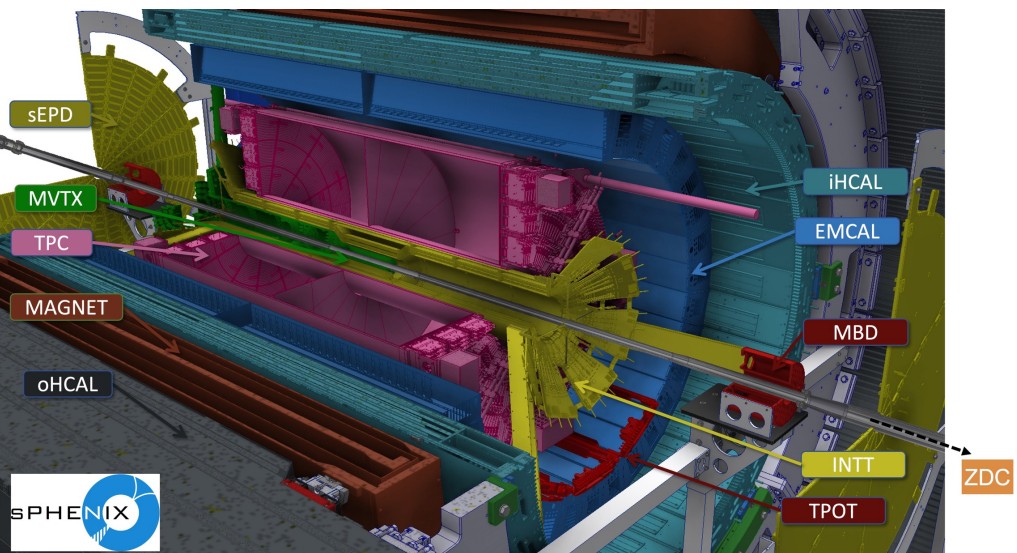

**Figure 1.** Schematic diagram of the sPHENIX detector, including all its subdetector systems.

The sPHENIX calorimeter system includes the electromagnetic calorimeter (EMCal) and hadronic calorimeter (HCal). Both the EMCAL and the HCal demonstrate excellent energy linearities and resolutions in Fermilab test beam experiments. Based on the hodoscope position-dependent correction, the tower averaged energy resolution of the EMCal is $\Delta E / \langle E \rangle = 3.5\% \oplus 13.3\% / \sqrt{E}$ [5]. The energy resolution of the HCal is $\Delta E / E = 11.8\% \oplus 81.8\% / \sqrt{E}$ [6]. Both of these meet the requirements for its physics goals. Moreover, sPHENIX is the first experiment with a full $2\pi$ azimuthal barrel hadronic calorimeter in the mid-rapidity range ($|\eta| < 1$) at RHIC, enabling full jet reconstruction in heavy-ion collisions.

## 2. The Monolithic Active Pixel Sensor-Based VerTeX (MVTX) Detector

MVTX is a silicon pixel detector with excellent position resolution approaching 5 μm and offers a continuous streaming readout option with a strobe length (a readout frame, here set to be 89 μs) as short as 5 μs [7], both of which are crucial for sPHENIX heavy flavor physics studies. It is adapted from the inner three layers of the inner tracking system-2 (ITS-2) detector from the ALICE experiment [8]. The sPHENIX MVTX consists of 48 staves (rectangular flat planes that hold silicon sensors along the beam axis), made of nine ALICE Pixel Detector chips [9], and electronic signals are readout by one front-end readout unit (RU). At the back end, the data will be transmitted to the Front-End LInk eXchange (FELIX) board for further processing [10]. One FELIX reads out eight RUs. Six FELIX systems are used to service the whole MVTX system.

In addition to the detector and readout electronic systems, we employ negative pressure cooling to maintain an overall constant temperature of both the silicon sensors and the readout electronics, for their normal operation and safety. Finally, slow control and quality control systems are used to operate and monitor the detector status, ensuring high-quality data taking.

MVTX staves were hand carried by air from the European Organization for Nuclear Research (CERN) to Lawrence Berkeley National Laboratory (LBNL). Two half barrels of the MVTX detectors were assembled at LBNL and then shipped to Brookhaven National Laboratory (BNL) in October 2022. In March 2023, we installed the MVTX cooling and electronic systems and successfully inserted the MVTX detector to the sPHENIX experiment. All 48 staves passed the electrical tests. We were able to readout the entire MVTX and collect fake hit rate and threshold scan data before the RHIC Au ion beam was turned on.

### 3. First Year Data Taking

sPHENIX anticipates collecting data from 2023 to 2025 [11]. In 2023, the Au + Au collisions at $\sqrt{s_{NN}} = 200$ GeV data taking had to be curtailed, due to an unfortunate valve box failure at RHIC on 1 August 2023. Nevertheless, from 18 May to 1 August 2023, sPHENIX has made significant progress in the detector commissioning with beam. All sPHENIX detector subsystems were able to take and record collision data. sPHENIX continued taking another two months of cosmic data after 1 August for alignment and calibration purposes.

Currently, we are carrying out data production and event reconstruction. Offline data analyses are also ongoing. As part of the tracking system, MVTX is able to synchronize with INTT, TPC, and TPOT to record cosmic ray events. A cosmic event display for tracking commissioning is shown in Figure 2.

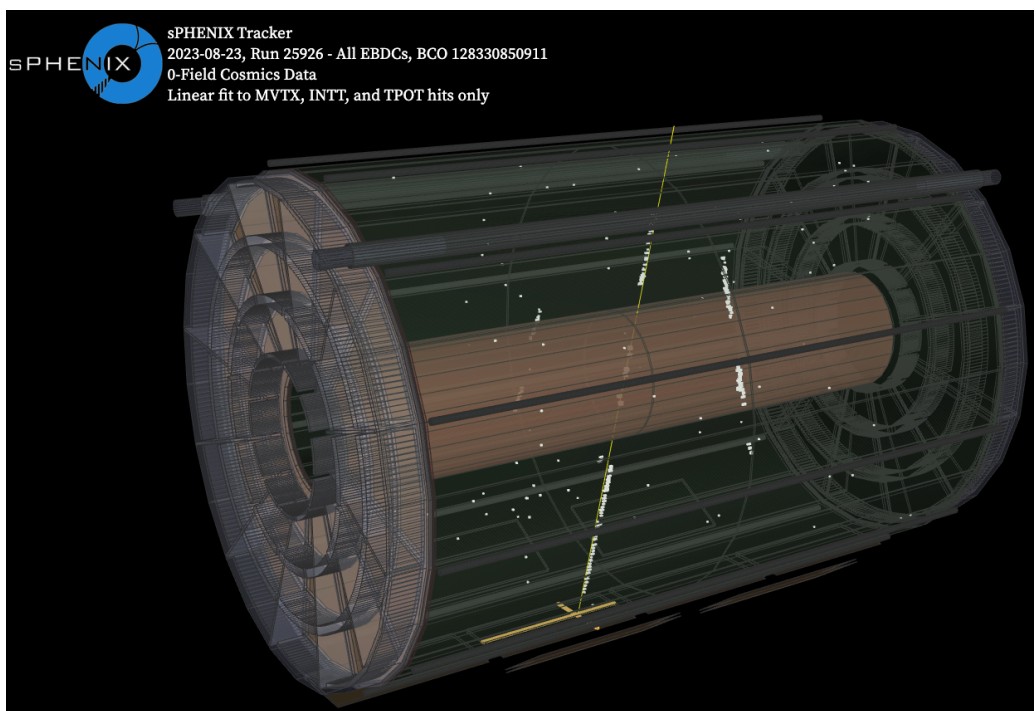

**Figure 2.** A cosmic event display of the tracking system taken on 23 August 2023. In this event, the Level 1 physics trigger for cosmic events, which requires the coincident signals between two outer HCal towers, is fired. In addition, the magnetic field is turned off (B = 0).

During the run, MVTX demonstrated a low noise level of less than $10^{-6}$ hot pixels per chip per event after masking hot pixels [12]. The MVTX mostly operated in continuous streaming readout mode, with a few runs in triggered mode. For this online display only, to reject the background and obtain pixels fired by cosmic muons, we remove clusters with single pixels. We can see a clear straight line cosmic muon track from the hits of MVTX, INTT, TPOC, and TPOT.

In addition, we analyzed the Au + Au collision data. We correlated the total number of pixels fired between two MVTX layers. In addition, we also correlated clusters of INTT and TPOT. Figure 3 shows the strong correlations from the subsystems.

The studies of both Au + Au beam collisions and the cosmic events demonstrate the functionality of the sPHENIX tracking system and its readiness for offline data analysis.

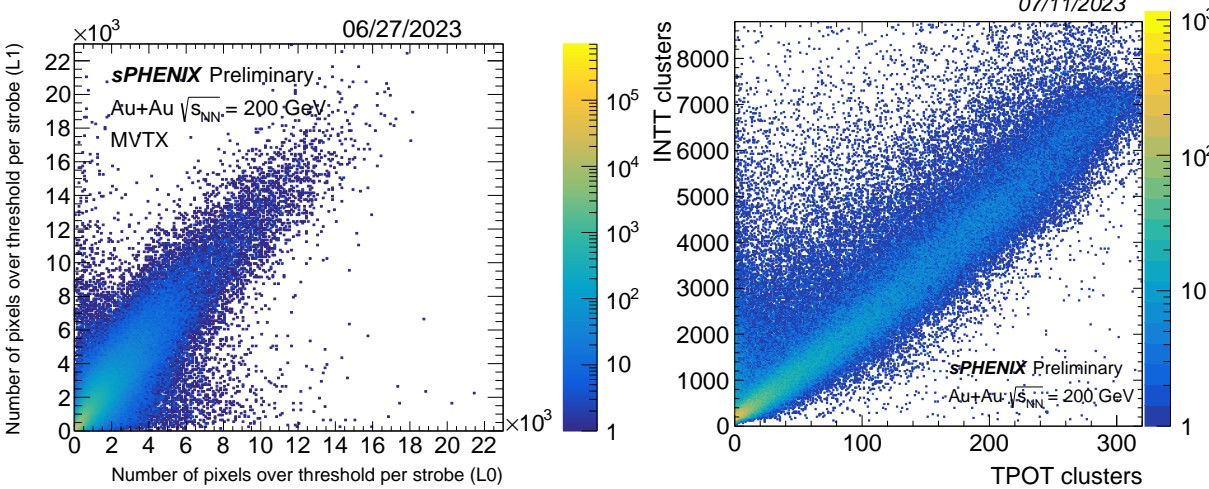

**Figure 3.** The correlation between MVTX layer 0 and 1 within one strobe is shown on the left, and INTT and TPOT clusters are shown on the right. Strong correlations are observed in MVTX, INTT, and TPOT, with beam collisions.

## 4. Heavy Flavor Physics Program

To complete the RHIC science mission, the sPHENIX physics program consists of a jet substructure, open heavy flavor, quarkonia, bulk physics, and cold QCD physics.

sPHENIX will take Au + Au data at $\sqrt{s_{NN}} = 200$ GeV in 2023, $p^{\uparrow} + p^{\uparrow}$ at $\sqrt{s} = 200$ GeV in 2024, and high luminosity Au + Au at $\sqrt{s_{NN}} = 200$ GeV in 2025. With excellent luminosity recorded, and tracking and vertexing capabilities, sPHENIX will be able to perform precise measurements of fully reconstructed open heavy flavor hadrons such as $D^0$, $B^+$, and $\Lambda_c^+$.

Thanks to their heavy masses, which are much greater than the QCD scale, $\Lambda_{QCD}$, and QGP temperature, $T_{QGP}$, ($m_Q \gg \Lambda_{QCD}$ and $m_Q \gg T_{QGP}$), heavy quarks, such as charm and beauty quarks, are produced in hard scattering processes in the early stage of heavy-ion collisions and have long thermal relaxation times. Their initial production spectra can be calculated by perturbative QCD. Heavy quarks retain their flavor and mass identities as they traverse through the QGP, making them excellent probes to study the transport properties of QGP. A heavy flavor particle widely used to study QGP is the $D^0$ meson, because of its large charm quark fragmentation fraction and simple hadronic final states in the decay channel of $D^0 \to K^- \pi^+$. Figure 4 shows fully reconstructed $D^0$ mesons in the decay channel of $D^0 \to K^- \pi^+$, with sPHENIX full detector simulations of Au + Au collisions.

According to the simulation studies shown in Figure 4, we expect to obtain the statistics that can observe clear $D^0$ resonance with about 25 min of data taking at 15 kHz. Thanks to the large minimum bias p + p and Au + Au datasets, excellent statistics can be achieved for precise and differential $D^0$ measurements. The projected performance of classic observables, such as the $D^0$ nuclear modification factor $R_{AA}$ and the elliptic flow $v_2$, which characterize the microscopic properties of the QGP, such as energy loss and transport coefficients, are shown in Figure 5.

With the precision vertex resolution, we can determine prompt and non-prompt $D^0$ meson using the distance of closest approach (DCA) in a data-driven manner. The studies of $R_{AA}$ and $v_2$ of prompt $D^0$ mesons will help us investigate charm quark thermalization in the QGP and interaction with the medium constituents. From the $R_{AA}$ and $v_2$ measurements of non-prompt $D^0$ feed down from b-hadrons, we can study the flavor dependence of energy loss and constrain beauty quark transport coefficients in the QGP.

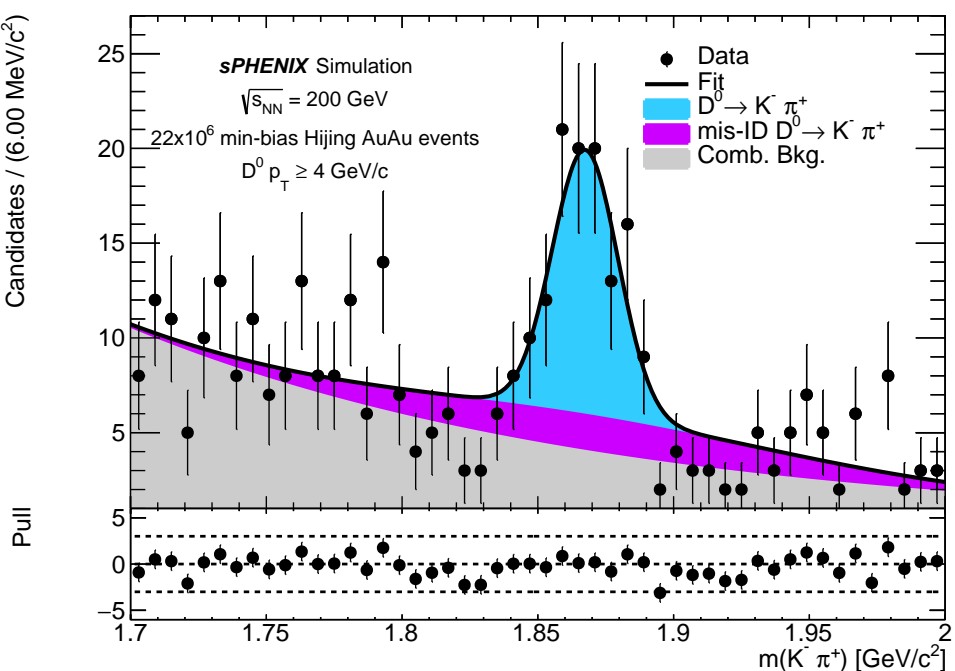

**Figure 4.** Invariant mass distribution of both prompt and non-prompt $D^0$ in the decay channel of $D^0 \rightarrow K^- \pi^+$ from HIJING Au + Au event generator with pile up with the sPHENIX detector in GEANT 4 simulations [13]. In the simulations, the KFParticle package was applied for secondary vertex reconstruction. It should also be noted that this constitutes 25 min of full luminosity data.

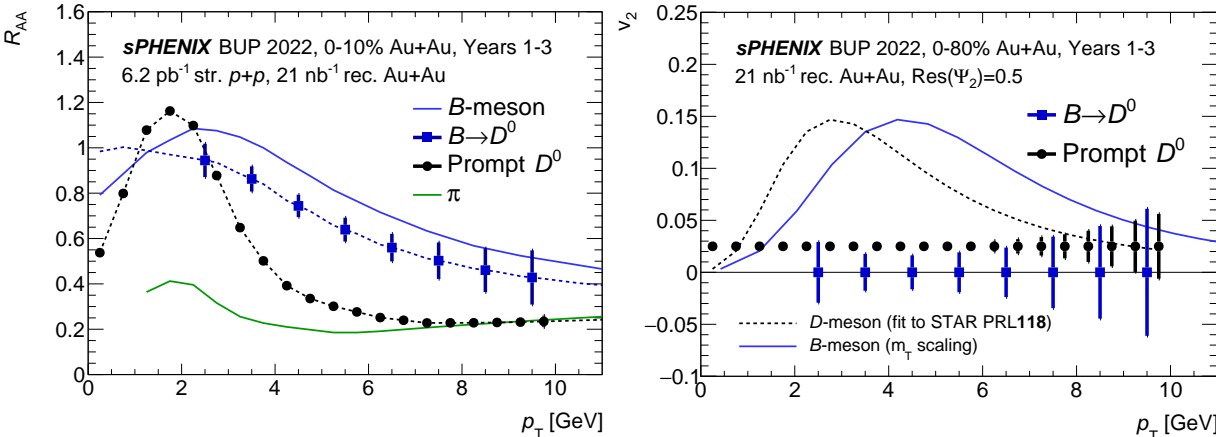

**Figure 5.** The projected performance of nuclear modification factor $R_{AA}$ (**left**) and elliptic flow $v_2$ (**right**) of prompt (black) and non-prompt (blue) $D^0$ as a function of $D^0$ $p_T$, with the sPHENIX experiment from simulations [11], are shown above.

The charm baryon $\Lambda_c^+$ has a more complex three-prong hadronic decay topology: $\Lambda_c^+ \rightarrow pK^- \pi^+$. sPHENIX is also able to fully reconstruct prompt $\Lambda_c^+$. Preliminary studies to tag protons using $dE/dx$ are underway, to improve $\Lambda_c^+$ reconstruction. Figure 6 shows the projected $\Lambda_c^+/D^0$ ratio performance with sPHENIX in p + p and Au + Au collisions.

The precise and differential measurement of $\Lambda_c^+/D^0$ with sPHENIX over a wide range of $p_T$ and event multiplicity or centrality from p + p to Au + Au will pinpoint the fragmentation and recombination mechanisms of charm quark hadronization from vacuum to QGP at RHIC.

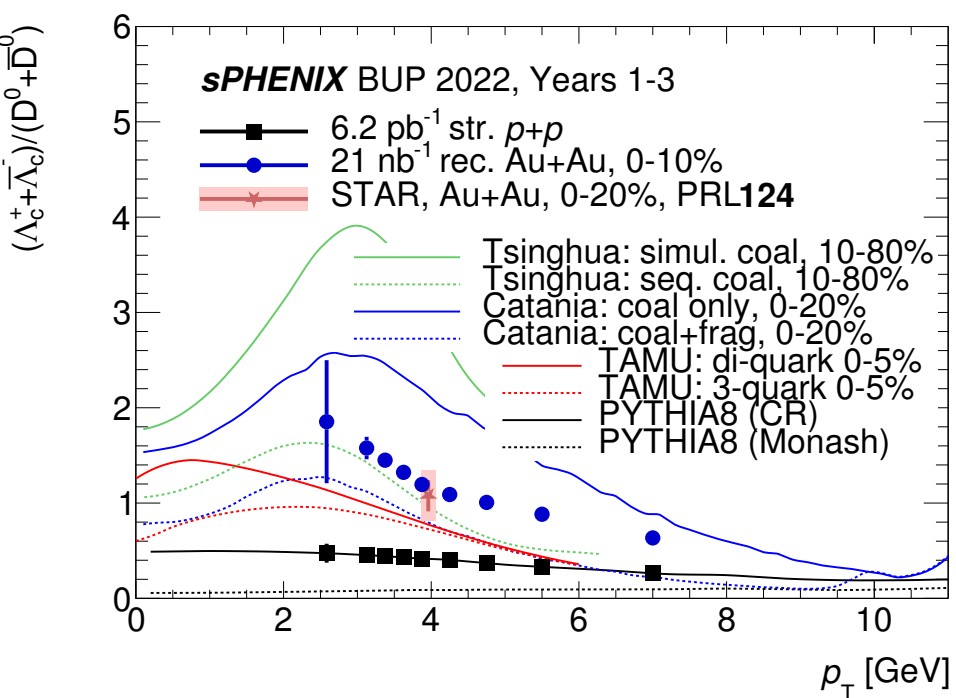

**Figure 6.** Projected performance of $\Lambda_c^+/D^0$ as a function of $\Lambda_c^+$ $p_T$ with full sPHENIX detector simulations in p + p (black) and Au + Au (blue) [11]. The data point of $\Lambda_c^+/D^0$ with STAR Au + Au data at 200 GeV (red) is presented [14]. Theoretical model predictions including Tsinghua, Catania, TAMU, and PYTHIA 8 are also overlaid.

Apart from $D^0$ and $\Lambda_c^+$, sPHENIX is capable of measuring fully reconstructed $D_s^+$ mesons to investigate charm hadronization at RHIC. Full sPHENIX detector simulations of fully reconstructed open b-hadrons, such as $B^+$, $B^0$, and $B_s^0$ in p + p and Au + Au collisions, aiming to study beauty quark diffusion, hadronization, and energy loss in the QGP, are currently ongoing. A dedicated discussion of sPHENIX b-physics projections can be found at [15].

In addition to fully reconstructed open heavy flavor hadrons, sPHENIX is capable of studying b-jets substructure and heavy flavor jet physics at RHIC. Figure 7 shows the $R_{AA}$ as a function $p_T$ and jet $z_g$ distributions of b-jet in p + p and Au + Au collisions.

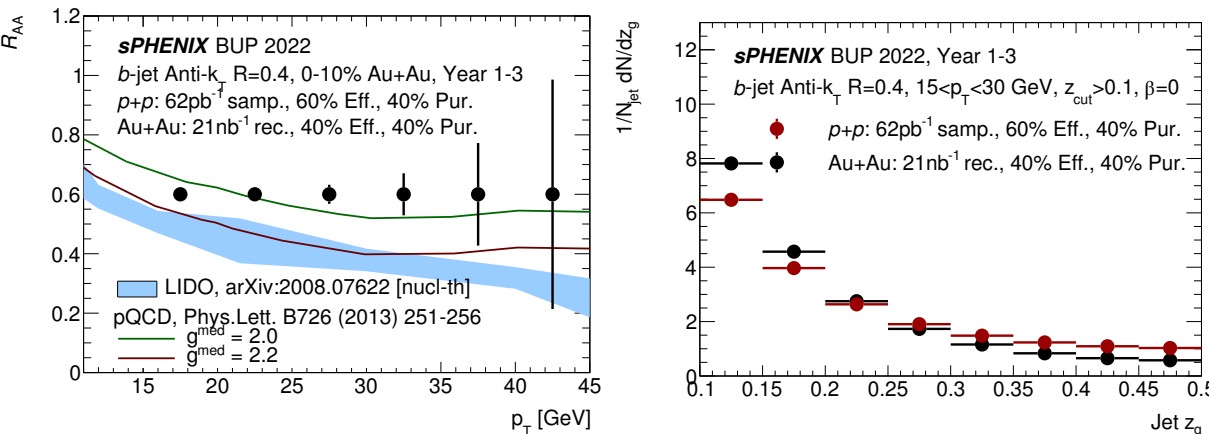

**Figure 7.** Projected precision of b-jet $R_{AA}$ as a function of $p_T$ (left) and substructure observable $\frac{1}{N_{jet}}\frac{dN}{dz_g}$ as a function jet $z_g$ in p + p and Au + Au with full sPHENIX detector simulations [11]. Wherein, LIDO (left) refers to [16] and pQCD (left) refers to [17].

Equipped with a high-performance hadronic calorimeter (HCal), sPHENIX is expected to deliver the first full b-jet measurements at RHIC. Moreover, sPHENIX will have excellent b-jet reconstruction and tagging capabilities thanks to the calorimeters and MVTX. sPHENIX will be able to perform differential subjet splitting function measurements with good precision at low $p_T$. The b-jet substructure measurements will test pQCD model calculations in p + p collisions and quantify the medium modification in the unique sPHENIX kinematic region in Au + Au collisions, complementary to LHC jet substructure measurements.

Finally, quarkonium spectroscopy stands as one of the flagship measurements of sPHENIX hidden heavy flavor physics. Because of the color screening effect, the binding energy of Y will decrease as the temperature of QGP increases [18]. In experiments, the sequential suppression of Y(1S), Y(2S), and Y(3S) production will be quantified by the nuclear modification factor $R_{AA}$ [19]. Hence, Y can be used as a thermometer to measure the temperature of QGP. The excellent performance of the sPHENIX EMCal and tracking detectors enables outstanding electron identification capabilities to perform precise $Y \rightarrow e^+ e^-$ measurements with sufficient invariant mass resolutions to distinguish all three Y states. Figure 8 shows the dielectron invariant mass distribution near the Y resonances in central Au + Au simulations and Y(1S), Y(2S), and Y(3S) $R_{AA}$ as a function of $p_T$.

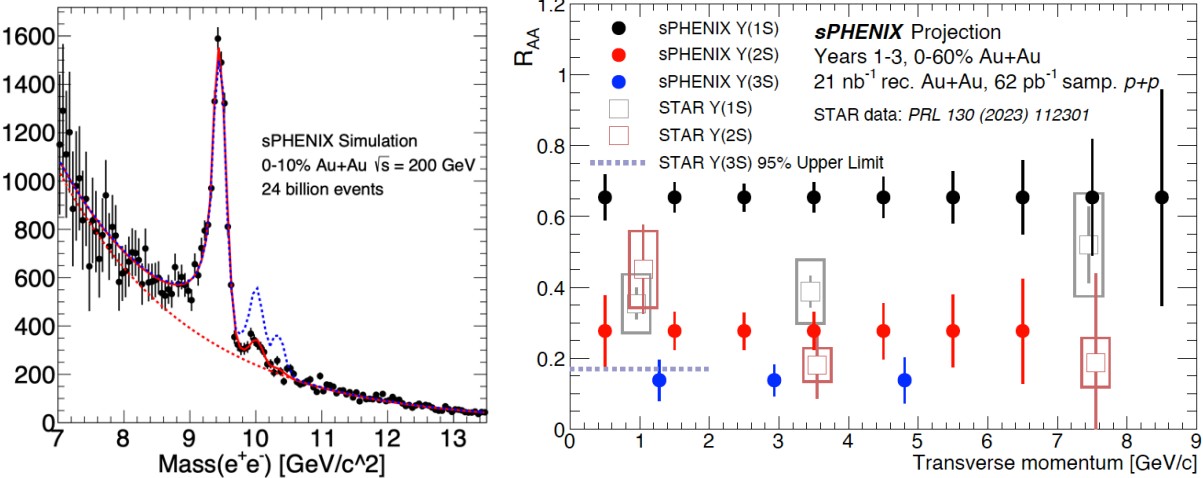

**Figure 8.** Projected performance Y(1S), Y(2S), and Y(3S) invariant mass distribution in 0–10% Au + Au collisions at $\sqrt{s_{NN}} = 200$ GeV from dielectron decay channel using sPHENIX simulation scaled up to 24 billion events is shown on the left. The Y(1S) (black solid circle), Y(2S) (red solid circle), and Y(3S) (blue solid circle) $R_{AA}$ as a function of $p_T$ of sPHENIX projection as well as the STAR data (open squares) [20] are shown on the right.

We can see that sPHENIX will be able to provide very precise and broad $p_T$ coverage (up to about 9 GeV/c) of Y $R_{AA}$ measurements for quarkonia physics studies. In addition, the Y(3S) resonance may potentially be separated from the Y(2S) and observed in Au + Au collisions for the first time by sPHENIX at RHIC, which is complementary to the recent observation and $R_{AA}$ measurements of Y(3S) with the CMS 2018 LHC PbPb data [21].

## 5. Summary

We have introduced the sPHENIX experiment, which is the first new experiment at RHIC in over 20 years to study the microscopic properties of QGP and is currently taking data. The sPHENIX physics program consists of jets, open heavy flavor, quarkonia, bulk physics, and cold QCD. In the 2023 Au + Au run, sPHENIX has demonstrated overall detector functionality and readiness for offline data analysis. The correlation studies of tracking and calorimeter systems validate the synchronization among subdetectors. Aside from the 200 GeV Au + Au collision data, sPHENIX has taken large cosmic datasets for detector alignment and calibration studies. In several cosmic ray events, a clear straight

line muon tracks are observed without the magnetic field, showing the overall good quality of the sPHENIX data.

We have also reported the projected physics measurements with full sPHENIX detector simulations, assuming the luminosity documented in the 2022 sPHENIX Beam Use Proposal [11]. We expect to achieve high statistics and fully reconstructed charm and beauty hadron measurements to study heavy quark diffusion, energy loss, and hadronization in QGP. Moreover, thanks to the excellent calorimeter, tracking, and vertexing performance, sPHENIX will carry out the first inclusive b-jet measurements with high precision at low $p_T$, complementary to LHC experiments. Finally, Y spectroscopy, one of the major physics topics of the sPHENIX experiment, provides us with precision measurement of QGP temperature. Potential observation and $R_{AA}$ measurements of Y(3S) for the first time may be accomplished by sPHENIX at RHIC.

**Funding:** This research was funded by Los Alamos National Laboratory, grant number 20220698PRD1; Knowledge Unlatched (Germany), grant number DE136766623.

**Data Availability Statement:** Data are contained within the article.

**Acknowledgments:** We would like to express our gratitude to the ISMD 2023 conference organizers for inviting us to present this work and providing useful information for the proceedings submission. We particularly want to thank Máté Csanád for his friendly correspondence and cordial reception. This work is supported by the United States Department of Energy Office of Science and Los Alamos National Laboratory Laboratory Directed Research & Development (LDRD).

**Conflicts of Interest:** The author declares no conflicts of interest.

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
