# Peer review of "Heavy Flavor Physics at the sPHENIX Experiment"

_universe, doi:10.3390/universe10030126_

Round 1

Reviewer 1 Report

Comments and Suggestions for Authors

I read this contribution with pleasure, it is clearly written, with just a few typos here and there

Let me mention:

line 75 comic -> cosmic

line 95 plans to take -> has taken

caption Fig.4 from with -> from

line 163 colorimeter -> calorimeter

line 164 have taken -> has taken

Comments on the Quality of English Language

It is fairly good. A further reading by a mother-tongue English writer may be of some help.

Author Response

Please find my response to referee 1 in the attachment.

Reviewer 2 Report

Comments and Suggestions for Authors

The Authors presented a well-written article, I only have a few comments for consideration:

- The article could be more explicit about the so-far unexplored kinematic region which the detector will be able to cover.
- When the continuous streaming mode is mentioned, it may be worth it to mention (or even compare it to) the continuous reading mode of ALICE, the LHC's dedicated heavy-ion detector.

- line 116 and 140: period is missing at the end of the sentence.
- line 163: "colorimeter" -> "calorimeter".
- I'm wondering, in Fig. 6, the PYTHIA CR simulation result (solid black line) was made by an extended color reconnection model? Which kind of model exactly?

Author Response

Please find my response to referee 2 in the attachment.

Reviewer 3 Report

Comments and Suggestions for Authors

This manuscript presents a very good introduction to the sPHENIX experiment, and reports ongoing and prospective physics measurements, especially the heavy flavor physics measurements that would be a major focus of the experiment, from the points of view of a dedicated experimental expert. Given the significance of the sPHENIX experiment to heavy-ion physics, the present manuscript will be undoubtedly of great interest to a broad readership. Therefore, it should be published as soon as possible. 

Before it is accepted for publication, I suggest the author discuss a bit more potential heavy flavor observables. For example, can Bs mesons be separated from B mesons? And what about Lambda_b and Bc mesons? The measurements of the full b-hadron family (and probably also their v2), in particular when pushed down to very low/zero pT, would be very interesting from the point of view of looking into the b-quark diffusion coefficient, hadronization mechanisms including the (differential) hadro-chemistry. 

In addition, the Ds-meson and Lambda_c measurements published already by STAR experiment suffered from significant error bars and kinematic limitations. It will be great if sPHENIX could improve on these measurements at RHIC energy. Fig.6 already touched upon Lambda_c (although the curves quoted there look pretty puzzling and dirty, and certainly do not represent the most convincing models and results), but the author may be willing to comment more on the Ds mesons.

By the way, I would like to inform that the talk “Exploring b-physics at sPHENIX” by C. Dean at the RBRC Workshop: Predictions for sPHENIX, https://indico.bnl.gov/event/15482/

indeed discussed more observables of b-hadrons.

Author Response

Please find my response to referee 3 in the attachment.

Round 2

Reviewer 2 Report

Comments and Suggestions for Authors

Dear Author,

thank you for addressing my comments and making improvements on the text. The manuscript is ready to be published.

Reviewer 3 Report

Comments and Suggestions for Authors

Thanks for the revisions. Now it should be published.